# Effect of Dietary Difructose Anhydride III Supplementation on the Metabolic Profile of Japanese Black Breeding Herds with Low-Level Chronic Exposure to Zearalenone in the Dietary Feed

**DOI:** 10.3390/toxins17080409

**Published:** 2025-08-14

**Authors:** Topas Wicaksono Priyo, Naoya Sasazaki, Katsuki Toda, Hiroshi Hasunuma, Daisaku Matsumoto, Emiko Kokushi, Seiichi Uno, Osamu Yamato, Takeshi Obi, Urara Shinya, Oky Setyo Widodo, Yasuho Taura, Tetsushi Ono, Masayasu Taniguchi, Mitsuhiro Takagi

**Affiliations:** 1Joint Graduate School of Veterinary Sciences, Yamaguchi University, Yamaguchi 753-8515, Japanmasa0810@yamaguchi-u.ac.jp (M.T.); 2Department of Reproduction and Obstetrics, Faculty of Veterinary Medicine, Universitas Gadjah Mada, Yogyakarta 55281, Indonesia; 3Shepherd Central Livestock Clinic, Kagoshima 899-1611, Japan; sasazaki.tribe@gmail.com (N.S.); t1643eson@yahoo.co.jp (K.T.); hasu@fa3.so-net.ne.jp (H.H.); papashepherd@gmail.com (D.M.); 4Faculty of Fisheries, Kagoshima University, Kagoshima 890-0056, Japan; kokushi@fish.kagoshima-u.ac.jp (E.K.); uno@fish.kagoshima-u.ac.jp (S.U.); 5Joint Faculty of Veterinary Medicine, Kagoshima University, Kagoshima 890-0062, Japan; osam@vet.kagoshima-u.ac.jp (O.Y.); k7307140@kadai.jp (T.O.); 6Soo Veterinary Clinic, Kagoshima A.M.A.A., Soo 899-8242, Japan; shinya-u@nosai46.jp; 7Division of Animal Husbandry, Faculty of Veterinary Medicine, Universitas Airlangga, Surabaya 60115, Indonesia; oky.widodo@fkh.unair.ac.id; 8Joint Faculty of Veterinary Medicine, Yamaguchi University, Yamaguchi 753-8515, Japan; ytaura@yamaguchi-u.ac.jp (Y.T.); yt-ono@yamaguchi-u.ac.jp (T.O.)

**Keywords:** Japanese Black cattle, difructose anhydride III, metabolic profile, zearalenone, intestinal barrier function

## Abstract

Mycotoxin contamination in animal feed can cause acute or chronic adverse effects on growth, productivity, and immune function in livestock. This study aimed to evaluate the impact of difructose anhydride III (DFA III) supplementation on serum biochemical parameters and intestinal environment in Japanese Black (JB) breeding cows under low-level chronic dietary exposure to zearalenone (ZEN). Using urinary ZEN concentration as an indicator of exposure, 25 JB cows were selected from a breeding farm with confirmed natural feed contamination. Blood samples were collected before DFA III supplementation (day 0), and on days 20 and 40 post-supplementation. Serum biochemical parameters and short-chain fatty acid concentrations were measured. During the studies, dietary ZEN concentration increased, yet improvements were observed in liver function, nutritional status, immune response, and inflammatory markers. Notably, serum butyrate concentration significantly increased following DFA III administration. These findings suggest that DFA III may positively influence intestinal microflora and enhance intestinal barrier function, which could contribute to improved health and nutritional status in cattle exposed to low-level chronic dietary ZEN contamination. DFA III supplementation may represent a promising strategy for mitigating the effects of low-level mycotoxin exposure in livestock production systems.

## 1. Introduction

Mycotoxin contamination in animal feed, caused by secondary metabolites from filamentous fungi such as *Fusarium* spp., is a persistent challenge in livestock production. These toxins, including zearalenone (ZEN), can induce acute or chronic disruptions in growth, reproduction, immune regulation, and metabolic function in animals [1,2]. To address this, there is a growing need for effective detection tools and nutrition strategies that mitigate the biological effects of dietary mycotoxins. We previously established a urinary concentration monitoring system for ZEN, a mycotoxin known for its endocrine-disrupting effects in animals. This system combines enzyme-linked immunosorbent assay (ELISA) screening with liquid chromatography–tandem mass spectrometry (LC–MS/MS) verification [3,4] and has been applied to breeding cattle herds: Japanese Black (JB) for beef and Holstein for dairy production.

While some mycotoxins are inactivated by rumen flora as the first barrier, others pass through the rumen, disrupt rumen flora, or are transformed into biologically active metabolites. Due to their antimicrobial activity, they can disrupt rumen flora and modulate the host immune system even at low doses [5]. Common adsorbent agents used to counteract these effects include mineral clays [6], yeast cell wall derivatives [7], enzymes, probiotics such as lactic acid bacteria [8], and prebiotics such as oligosaccharides [9].

Prebiotics are specific substrates utilized by host microbes to confer health benefits [10]. Oligosaccharides have been shown to interact directly with intestinal epithelial cells, enhancing intestinal barrier function and modulating immunological responses [11]. Non-digestible oligosaccharides, such as mannan oligosaccharides [12], fructooligosaccharides [13], and lactulose, have attracted interest for their potential to reduce disease incidence in animals by promoting beneficial gut flora by increasing the populations of *Bifidobacterium* spp. [14]. Short-chain fatty acids (SCFAs), such as acetic, propionic, and butyric acids, are produced through microbial fermentation of non-digestible carbohydrates and are known to be stimulated by prebiotic intake [15]. SCFAs contribute to improved gut microbiota composition [15], reduce inflammation [16], and inhibit mycotoxin absorption [17]. SCFAs are products of non-digestible carbohydrate fermentation by anaerobic gastrointestinal bacteria. They are also produced endogenously through metabolism in body tissues and ingested through the diet [18]. Proper microbiota colonization in the gastrointestinal system and SCFAs may play a vital role in regulating inflammatory response [18,19], maintaining intestinal barrier integrity [20], and serving as indicators of host health status [21]. In light of increasing concerns about antimicrobial resistance due to the overuse of antibiotics in livestock, alternatives such as probiotics and prebiotics are gaining prominence for their health-promoting properties.

Difructose anhydride III (DFA III), or di-D-fructofuranose-1,2′:2,3′-dianhydride, is a non-digestible disaccharide derived from inulin via microbial fermentation of chicory root [22]. Previous studies have shown that DFA III enhances paracellular calcium (Ca) absorption in cattle [23]. In earlier work, we demonstrated that oral DFA III supplementation in JB calves, used as a prebiotic, improved gut flora composition and overall health status [24,25]. We have also reported that our urinary ZEN concentration monitoring system was used to estimate ZEN contamination levels in feed and to track intestinal ZEN adsorption following DFA III supplementation [26]. With the serum samples derived from the same herd [26], we recently reported that DFA III administration modulated blood SCFA concentrations in JB heifers, suggesting improvements in the intestinal environment and barrier function [27].

Based on these findings, we hypothesized that DFA III supplementation could mitigate the adverse effects of mycotoxin contamination, improve the intestinal barrier integrity, and reduce systemic inflammation in cattle. To our knowledge, no studies have simultaneously assessed the effect of dietary DFA III supplementation on mycotoxin contamination, inflammatory biomarkers, serum biochemistry, and circulating SCFA concentrations in JB herds. This study aimed to examine the effects of DFA III supplementation on serum biochemical parameters such as anti-Müllerian hormone (AMH), which represents antral follicle reserve in the ovary, and serum amyloid A (SAA), which is an acute-phase protein and nutritional indicator in JB cattle with chronic dietary ZEN contamination. We also investigated changes in the intestinal environment by measuring serum SCFA concentration before and after DFA III supplementation to clarify its effects on intestinal barrier function under chronic mycotoxin contamination.

## 2. Results

### 2.1. Study of Metabolic Profiles in Preliminary ZEN Contamination Screening

The urinary ZEN/creatinine (Cre) concentrations (pg/mL Cre, mean ± SEM) for Herds 1, 2, and 3 were 749.3 ± 19.9, 1885.2 ± 578.6, and 4575.3 ± 927.2, respectively. All metabolic profile parameters were within the normal reference ranges, except for AMH and SAA, for which no established reference ranges exist. Significant differences were observed in free fatty acid (FFA) (*p* = 0.000), blood urea nitrogen (BUN) (*p* = 0.027), glucose (Glu) (*p* = 0.002), triglyceride (TG) (*p* = 0.039), 3-hydroxybutyrate (3HB) (*p* = 0.000), and total protein (TP) (*p* = 0.034), indicating variability in nutritional status. The albumin/globulin (A/G) ratio also differed significantly (*p* = 0.013), suggesting disparities in nutritional and immune status, particularly malnutrition in Herd 3. A significant difference in serum magnesium (Mg) concentrations (*p* = 0.022) was also noted, possibly due to differences in mineral supplementation or roughage composition among herds. No significant differences were observed in AMH, SAA, glutamic oxaloacetic transaminase (GOT), gamma-glutamyl transferase (GGT), total cholesterol (T-Cho), Ca, inorganic phosphate (IP), vitamin A (Vit. A), and albumin (Alb). Based on these results, a recommendation was made to slightly increase protein and energy intake in the diet of Herd 3. Preliminary screening results exploring the relationship between ZEN contamination and serum biochemical parameters in three JB herds are shown in Appendix A.

A follow-up screening conducted in May 2022 (approximately 6 months later) for Herd 3 (Appendix A) showed no change in urinary ZEN concentration, confirming persistent ZEN contamination. However, significant differences in the indicators of nutritional status, such as FFA, BUN, Glu, 3HB, and TP concentrations, were confirmed. This indicated an improvement in the nutritional status of the herd.

Re-monitoring revealed no change in urinary ZEN concentration, and chronic ZEN exposure at the same levels was confirmed during the two sampling periods. Therefore, we conducted a DFA III supplementation trial for Herd 3.

### 2.2. Effects of DFA III Supplementation

This study assessed the effects of DFA III supplementation on serum biochemistry and SCFA concentrations under conditions of chronic low-concentration ZEN contamination. Twenty-five cows from Herd 3 were included in the time-series analysis. Figure 1a shows the changes in urinary ZEN concentrations in the same two cows, while Figure 1b shows mean (±SEM) AMH and SAA concentrations during the period of DFA III supplementation. A statistically significant time-series change was observed in the AMH concentrations, with a significant decrease on days 20 and 40 relative to those of day 0. The change rates were 0.75 (95% CI: 0.71, 0.80; *p* < 0.001) on day 20 and 0.80 (95% CI: 0.75, 0.84; *p* < 0.001) on day 40 relative to day 0. No significant changes were observed in SAA concentrations during the supplementation period. Mean serum biochemical values are presented in Figure 2.

### 2.3. Parameters for Liver Function

GOT levels significantly increased on day 20 relative to day 0 (*p* < 0.001) and decreased by day 40 (*p* < 0.001), with no significant difference between the concentrations on days 0 and 40. The GGT concentrations for days 0 and 20 and days 20 and 40 were not significantly different, but a significant increase from day 0 to 40 was observed (*p* = 0.042).

### 2.4. Parameters for Nutritional Status

Several parameters, such as FFA, T-Cho, Glu, 3HB, and TP concentrations, demonstrated peak- or trough-shaped transitions from day 0 to day 40. FFA levels decreased on day 20 relative to day 0 (*p* < 0.001), increased by day 40 (*p* < 0.001), but remained statistically unchanged between days 0 and 40. The T-Cho concentration significantly decreased on day 20 relative to that of day 0 (*p* < 0.001), increased from day 20 to day 40 (*p* < 0.001), and significantly increased from day 0 to day 40. Glu and TP showed similar trends. In contrast, the 3HB concentrations significantly increased on day 20 relative to day 0 (*p* < 0.001) and remained elevated on day 40 (*p* = 0.034). BUN concentrations did not differ significantly between days 0 and 20 but increased significantly from day 20 to day 40 (*p* < 0.001) and from day 0 to day 40 (*p* < 0.001).

### 2.5. Parameters for Mineral Status

The Ca concentrations significantly increased on day 20 relative to the values on day 0 (*p =* 0.004), and no statistically significant changes were observed until day 40. IP levels significantly decreased from day 0 to day 20 (*p* < 0.001), increased from day 20 to day 40 (*p* < 0.001), and were overall higher from day 0 to day 40 (*p* = 0.042). The Mg concentrations significantly increased from day 0 to day 20 (*p* = 0.029) and remained elevated until day 40 (*p* = 0.037).

### 2.6. Parameters for Immune and Inflammation Status

Alb concentrations significantly decreased on day 20 compared to day 0 (*p* < 0.001) but significantly increased again by day 40 (*p* < 0.001), returning to baseline levels. The A/G ratio increased on day 20 relative to that of day 0 (*p* = 0.007), remained stable through day 40, and was significantly higher overall from day 0 to day 40 (*p* < 0.001).

### 2.7. Parameters for Vitamin Status

Vit. A level significantly decreased on days 20 and 40 compared with that on day 0 (*p* < 0.001), with further decline between days 20 and 40 (*p* = 0.007). Overall, Vit. A level declined significantly from day 0 to day 40 (*p* < 0.001). Vitamin E (Vit. E) concentration decreased on day 20 relative to that of day 0 (*p* < 0.001) but significantly increased between days 20 and 40 to a value close to that of day 0 (*p* < 0.001). Time-series trends of all biochemical parameters (linear mixed-model analysis) are presented in Appendix A.

### 2.8. Parameters for SCFA Status

Figure 3 presents SCFA trends over time, and Appendix A presents representative chromatograms on gas chromatography–mass spectrometry (GC/MS) of the SCFA standards analyzed in the present study. No time-series changes were observed in acetic acid and valeric acid concentrations after DFA III supplementation. Propionic acid, isobutyric acid, and isocaproic acid levels decreased significantly on days 20 and 40 relative to those of day 0 (*p* < 0.001). Butyric acid concentrations dropped significantly on day 20 relative to those of day 0 (*p* = 0.011), then returned to near baseline by day 40 (*p* < 0.001). 2-methylvaleric acid significantly decreased from day 20 to day 40 relative to those of day 0 (*p* < 0.001). Hexanoic (Caproic) acid concentrations significantly decreased only on day 40 relative to those of day 20 (*p* = 0.018). 2-Methylhexanoic acid and heptanoic acid significantly decreased only on day 40 relative to those of day 0 (*p* < 0.001). Heptanoic acid concentrations decreased significantly from day 0 to 20 (*p* < 0.001) and from day 20 to 40 (*p* = 0.005). Full SCFA analysis is provided in Appendix A.

## 3. Discussion

DFA III supplementation in cattle has traditionally been implemented post-parturition to prevent milk fever and improve productivity by promoting Ca and Mg absorption from the intestine. It has been shown to be safe and effective, and it is also added to the colostrum for newborn calves to support intestinal IgG absorption. Based on previous findings from dairy cattle, the present study aimed to verify the effects of DFA III supplementation in JB cattle. DFA III has been shown to reduce diseases in JB calves when added to milk replacements, and it can also inhibit the absorption of mycotoxins such as ZEN and sterigmatocystin (STC) when administered to JB heifers’ feed [26,28]. Based on this, this pilot study was conducted to extend investigations into the efficacy of DFA III supplementation in breeding cattle and was a field trial to verify the effect of DFA III supplementation in JB breeding cows, focusing on metabolic profile testing via blood biochemical analysis. A preliminary evaluation included blood biochemical screening and urinary ZEN monitoring in three JB breeding herds under a similar feeding regimen [3], identifying a chronically contaminated herd for further study.

The present study revealed significant differences in the concentrations of minerals, such as Ca, Mg, and IP, in JB cattle. Concentrations of all minerals increased from days 0 to 20 or from days 0 to 40, suggesting an increased absorption rate of minerals during DFA III supplementation. Several effects of oral DFA III administration, including increased intestinal Ca absorption [29], Mg absorption [30], and increased hemoglobin and iron concentrations [31], have been reported in humans, cows, and rats. Supplementation with DFA III has been linked to improved health in JB calves based on serum biochemical parameters [24,25] and reduced the urinary concentrations of ZEN and STC in JB heifers [26,28]. Mechanistically, DFA III may facilitate paracellular transport by decreasing transepithelial electrical resistance and improving paracellular marker movement, with changes in Caco-2 cells’ claudin-1, an element of tight junctions, and actin filaments [32,33]. This suggests that DFA III contributes to intestinal barrier integrity [34], as corroborated by the mineral profile improvements observed here. Even though there was no control group in this study, the observed alterations happened without any change in feed composition, which supports the conclusion that the results show DFA III effects.

Metabolic evaluation revealed significant differences in all serum biochemical parameters, except for TG concentrations. Liver function markers, such as GOT and GGT, were significantly different, primarily because GOT concentration decreased. Additionally, there were significant differences in nutritional conditions, such as FFA, T-Cho, Glu, Vit. A, Vit. E, TP, and Alb, with most parameters increasing within 40 days following DFA III supplementation. Previous studies confirm that 1–2 months of DFA III administration improves health and nutritional conditions of calves [24,25,35]. This indicates that DFA III supplementation as a prebiotic helps restore liver function and improve the nutritional status in JB cows contaminated with ZEN (during the supplementation test period, when urine ZEN concentrations increased). In this study, the urinary ZEN concentration was measured three times during the 40 days of supplementation. Despite unchanged feed, the urinary ZEN concentration tripled on average in the second and third samples compared to the sample collected when the supplementation started. This indicated an increase in the degree of ZEN infiltration in the diet. The results of each biochemical test during that period seemed to reflect the urinary ZEN concentrations. GGT, an indicator of liver function, significantly increased. Contrastingly, the FFA, T-Cho, BUN, Glu, and TP concentrations, which are indicators of nutritional status, significantly decreased, while 3HB concentrations significantly increased. However, the above-mentioned effects were alleviated from 20 to 40 days after the start of DFA III supplementation. A significant decrease in GGT and improvement in various nutritional status indicators were also confirmed, along with a significant decrease in 3HB concentration, which is an indicator of malnutrition. Furthermore, a significant increase in the A/G ratio, an indicator of immune and inflammatory status, was observed within 40 days of treatment. Compared with the results of a preliminary study involving three herds with different urinary ZEN concentrations, the results of this study did not reveal an effect of DFA III supplementation on the inhibition of ZEN absorption from the intestine, as indicated by the urinary ZEN concentrations. However, the results of several serum biochemistry tests suggest improved nutritional and inflammatory statuses of the herds.

Previous findings from DFA III administered for 2 weeks (ZEN concentrations in roughage and concentrate were 0.27 and 0.22 mg/kg, respectively) in 9- to 10-month-old JB cattle demonstrated reduced urinary ZEN and zearalenol (ZOL) concentrations [26]. LC–MS/MS on the day prior to DFA III administration, on days 9 and 14 during supplementation, and again 9 days after its discontinuation, revealed lower ZEN metabolites in the DFA III group and improved IP concentration on the 23rd day (8.4 versus 7.7 mg/dL). From these results, we report that DFA III reduces the concentration of mycotoxins that reach the systemic circulation and are excreted in urine and suggest that this preventive effect may be related to the improvement of tight junction-dependent intestinal barrier function [26]. The following findings were emphasized by a comprehensive reexamination of the previously described studies [26]. The effect of DFA III on urinary ZEN concentration was observed approximately 10 days after the start of DFA III dietary supplementation and persisted up to 9 days after the 2-week administration period was discontinued. No significant difference was observed between the urinary ZEN concentration and total ZEN concentration; however, a significant difference was confirmed in the concentrations of the urinary ZEN metabolites α-ZOL and β-ZOL. These results suggest that some degree of intestinal barrier function was maintained even after DFA III administration was discontinued, although the effect declined over the following 9 days.

AMH concentrations declined significantly between days 20 and 40, consistent with prior findings on the impact of ZEN on ovarian granulosa cells of pre-antral follicles and its influence on antral follicle counts [3]. Mycotoxin contamination, particularly ZEN and its metabolites, is known to disturb the antral follicle, interrupt the endogenous estrogenic response during the pre-ovulatory stage, decrease ovarian follicle maturation, and impair the steroidogenic function of ovarian granulosa cells [36]. In our previous study, we found that natural ZEN contamination may negatively correlate with AMH concentration in JB breeding cattle [3,4]. In the current study, urinary ZEN concentrations increased by day 20, coinciding with a reduction in AMH levels from day 0 to 20, followed by stabilization through day 40. These results support our previous findings regarding the relationship between urinary ZEN and AMH concentrations [3]. However, previous data indicate that AMH concentrations require approximately 2 months to recover in response to ZEN contamination [3], and the present results suggest that 30-day DFA III supplementation may not be sufficient to restore AMH concentrations affected by ZEN contamination. On the other hand, based on the results of the present study alone, it remains unclear whether the intestinal barrier protection conferred by DFA III helped mitigate the effects of ZEN exposure. It was speculated that the intestinal barrier function induced by DFA III supplementation may have been exerted after the inflammatory state of the intestinal epithelial cells was improved. Thus, it is possible that a delayed effect may occur following the initiation of DFA III supplementation. Further research is needed to elucidate the mechanism underlying the effect of DFA III supplementation.

No significant differences in SAA concentrations were observed during the 1 month of the DFA III supplementation period. The gastrointestinal microbiota plays a key role in immune system development and function, modulating both local and systemic inflammatory responses through interactions with epithelial cells, intestinal permeability, dendritic cells, and T and B immune cells [37]. Mycotoxin co-contamination, particularly ZEN, has been shown to induce an inflammatory response in the intestines [38,39] and liver [40] as well as disrupt immunity [41,42]. Inulin supplementation has been reported to mitigate both intestinal and systemic inflammation by increasing the production of SCFAs, especially butyrate [37]. Our previous findings also suggested that supplementation with DFA III may affect intestinal SCFA concentrations and improve intestinal barrier function in cattle [27]. Notably, SAA served as an effective marker of inflammation in highly ZEN-contaminated herds (urinary ZEN/Cre > 20,000 pg/mL Cre) in prior studies [3,4]. In contrast, the current study involved cattle exposed to relatively lower ZEN concentrations than in a previous report [3], suggesting that there may be no change in the SAA concentration after DFA III supplementation. Therefore, SAA may not be a sensitive indicator for monitoring inflammation, especially in the setting of chronic infiltration with low concentrations of ZEN. Interestingly, the concentration of Alb, another marker associated with inflammation, increased following DFA III supplementation. Therefore, the results of this study showed that DFA III supplementation may improve inflammation in vivo in a chronic, low-concentration, ZEN-contaminated environment.

In the present study, serum SCFA concentrations increased on day 20 and declined by day 40 following DFA III supplementation. This pattern most likely results from DFA III’s fermentation in the gastrointestinal tract and its subsequent absorption into the bloodstream. In cows, ruminal microorganisms have difficulty digesting DFA III. Studies of microflora in the rumen have indicated its low fermentability and degradability, which strongly implies that it passes through the rumen and enters the intestinal tract [23], especially the duodenum, 1 h after oral ingestion [43]. SCFAs, including acetate, propionate, and butyrate, are metabolites produced during bacterial fermentation of dietary fiber in the intestinal tract [44] and are known to regulate and maintain the integrity of epithelial intestinal barrier function and prevent inflammation and systemic immune responses via evolutionarily conserved pathways, including G protein-coupled receptor signaling and histone deacetylase inhibition [45]. The anti-inflammatory role of butyrate is mediated through its direct effects on the differentiation of intestinal epithelial cells, phagocytes, B cells, and plasma cells, as well as regulatory and effector T cells. Intestinally derived SCFAs also directly and indirectly affect immunity at extraintestinal sites, such as the liver, lungs, reproductive tract, and brain, and have been implicated in various disorders, including infections and intestinal inflammation [45]. In our previous report, DFA III supplementation altered serum SCFA profiles, significantly increasing isobutyric acid and decreasing valeric acid concentrations before and 2 weeks after supplementation [27]. In the present study, butyric acid concentrations decreased on day 20 of supplementation but significantly increased by day 40. Similar findings were reported by [29], who observed no significant differences in lactic acid, acetic acid, propionic acid, and butyric acid concentrations between the control and DFA III groups. Likewise, isobutyric acid levels in ruminal fluids remained constant upon supplementation with inulin, which is the source of DFA III [21]. Isobutyric acid, derived from the fermentation of amino acids, can stimulate the growth of amylolytic bacteria that prefer to use peptides and amino acids because of their proteolytic activity [46]. Additionally, the rumen increases the proliferation of cellulolytic bacteria, which is another reason for the decline in acetic acid concentrations [47]. Alternatively, DFA III supplementation did not increase specific SCFAs or other organic acids, suggesting a relatively low energy for growth in rats [48]. Additionally, DFA III supplementation did not increase the total SCFA level but increased the anaerobic bacteria population inside of cecum in rats [29]. Therefore, the results obtained in this study can be considered comparable to previous reports on DFA III supplementation tests [27], and the difference in the increase or decrease in SCFA concentration observed in some cases may be due to the difference in the feed given. Future research should investigate shifts in gut microbiota composition and population to better understand the fermentation behavior and metagenomics of DFA III in the intestinal tract.

There are certain limitations to this study. Due to the herd’s feeding management style, a control group without DFA III supplementation could not be included, potentially limiting the ability to definitively attribute observed effects of DFA III supplementation. However, the significant increase in mineral concentrations was considered an indirect indicator of the effects of DFA III supplementation, supported by findings from two preliminary metabolite profile tests in three JB breeding herds. In this study, statistically significant differences were observed in several parameters, including biochemistry results and SCFA concentrations, after DFA III supplementation within 40 days. Nonetheless, further studies incorporating a control group and correlation analysis of ZEN concentrations, biochemistry results, and SCFA profiles are warranted to enhance the robustness and interpretability of these findings.

## 4. Conclusions

In conclusion, a 40-day DFA III supplementation in cattle herds chronically exposed to low-level ZEN contamination may improve liver function, enhance nutritional status and immune response, and influence blood SCFA concentration trends. DFA III, as a prebiotic, has beneficial effects on the integrity of the intestinal barrier and overall intestinal health. These findings suggest that DFA III supplementation could help mitigate the adverse effects of dietary ZEN contamination. Further accumulation and integration of field data are warranted to establish this strategy as a viable mycotoxin control approach in livestock farming.

## 5. Materials and Methods

### 5.1. Ethical Approval

The guidelines and rules for the protection of experimental animals at Yamaguchi University in Yamaguchi, Japan (No. 110, 11 November 2008) were followed during the conduct of this study. All farmers who participated in the study provided informed consent.

### 5.2. Chemicals and Solvents

DFA III was donated by Nippon Beet Sugar Manufacturing Co., Ltd. (Obihiro, Japan). ZEN was purchased from MP Biomedicals (Heidelberg, Germany). A stock solution of ZEN (1 µL in methanol) was stored in a dark area at 4 °C. For the measurement of SCFAs using GC/MS, a mixture of acids was prepared with slight modifications based on a previous study [27]. The mixture included formic acid, acetic acid, propionic acid, isobutyric acid, butyric acid, 2-methylvaleric acid, isovaleric acid, valeric acid, isocaproic acid, hexanoic acid (caproic acid), 2-methylhexanoic acid, and heptanoic acid (Sigma Aldrich, Tokyo, Japan).

### 5.3. Preliminary ZEN Contamination Screening

In December 2021, after the rice harvest, preliminary screening was conducted at a JB cattle breeding farm to evaluate ZEN contamination in dietary roughage (rice straw and/or whole crop silage). Urinary ZEN levels were monitored in three neighboring herds (1, 2, and 3) located in the Kyushu region of Japan. A veterinarian provided routine care and consultation to the farmers. Under consistent management practices, all animals were housed indoors and fed concentrate and roughage separately. Sampling dates were coordinated across herds. Urine samples were collected from two cows with similar body weights in each herd via natural urination after gentle perineal massage, following a previously described method [3]. According to a previous study [3,4], two samples per herd were considered sufficient for estimating feed contamination levels. All concentrates administered to the cattle were commercially sourced and examined for mycotoxin contamination during manufacturing. Collected urine and blood samples were immediately cooled, protected from light, centrifuged, and frozen. The frozen samples were then transported to the laboratory and stored at −30 °C for ZEN and Cre analysis. Among the three herds, Herd 3 showed relatively higher urinary ZEN concentrations than Herds 1 and 2. Therefore, it was re-sampled in May 2022 for further ZEN and blood biochemistry analysis.

### 5.4. Animals and Management

The study was conducted between November and December 2022 using Herd 3 (female JB cattle). A total of 25 JB cows (mean age ± SEM, 6.2 ± 0.8 years; approximate body weight range, 400–600 kg) were enrolled. Ambient temperatures during the study period ranged from 7 to 21 °C. DFA III was administered at a dose of 40 g/day (20 g per feeding, twice daily), mixed with concentrate. The dosage used was based on a previous study [27]. All cows were fed a mixture of wrapped sorghum silage and barnyard millet as roughage, harvested from their herd’s pastureland, and supplemented with commercial concentrate formulated for JB breeding cows (Kurupita, Minami Hihon Kumiai Siryo Co., Ltd., Kagoshima, Japan). The concentrate contained digestive tract materials (46%), grain (32%), vegetable oil cakes (8%), and other components (14%), and was administered twice daily. Its nutritional composition included total digestible nutrients > 71.5%, crude protein > 15%, crude fat > 3%, crude fiber > 10%, crude ash > 10%, Ca > 0.8%, and phosphorus > 0.4%. The herd had a documented history of poor reproductive performance. Before starting the experiment, ovarian status was assessed, and ZEN levels were measured in urine samples from four cows (two per test group) using ELISA. The mean urinary ZEN concentrations (pg/mg Cre, ±SEM) were 4575.3 ± 927.2 and 4733.8 ± 42.3, respectively, confirming chronic low-level ZEN contamination in feed. This herd was thus selected as a model to evaluate the effects of DFA III supplementation on urinary ZEN, AMH, SAA, and blood biochemical parameters. A schematic diagram outlining the study design, including the preliminary screening process, is shown in Figure 4.

### 5.5. Urine and Blood Sample Collection

During the DFA III supplementation period, both urine (from two identical cows) and blood samples (*n* = 25) were collected at three time points: day 0 (prior to DFA III supplementation), day 20 (after 20 days of supplementation), and day 40 (10 days post-supplementation). Blood samples were collected from the jugular vein using a 10 mL plastic syringe, while urine samples (approximately 10 mL) were collected via perineal massage into 15 mL tubes. All samples were immediately placed in an ice-filled container to maintain low temperature and shielded from light during transport to the laboratory. Blood samples were centrifuged at 1409× *g* for 10 min. The resulting serum was divided and stored at −80 °C for SCFA analysis and at −30 °C for other biochemical tests. Urine samples were centrifuged at 1917× *g* for 5 min, transferred to 1.5 mL microtubes, and stored at −30 °C until analysis.

### 5.6. Measurement of ZEN, AMH, and SAA

Urinary ZEN was measured using a commercially available kit (RIDASCREEN^®^ Zearalenon; R-Biopharm AG, Darmstadt, Germany) with minor modifications to the manufacturer’s protocol. Briefly, 10 μL of β-glucuronidase/arylsulfatase solution was added to 3 mL sodium acetate buffer (50 mM, pH 4.8) containing 0.1 mL of five-fold diluted urine. After 15 h incubation at 38 °C, the mixture was purified using C18 solid-phase extraction (SPE) columns (Strata; Phenomenex, Torrance, CA, USA) preconditioned with 3 mL of methanol followed by 2 mL of 20 mm Tris buffer (pH 8.5)/methanol (80:20). The column was then washed with an additional 2 mL of Tris buffer (pH 8.5)/methanol (80:20) and 3 mL of methanol (40%), followed by centrifugation for 10 min at 39× *g* to dry the column. Elution was performed slowly (15 drops/min) with 1 mL of methanol (80%). The eluate was evaporated at 60 °C using a centrifugal evaporator and redissolved in 50 μL methanol. Subsequently, 450 μL of sample dilution buffer was added, thoroughly mixed, and an aliquot of 50 μL was used for the ELISA assay. Absorbance was measured at 450 nm using a microplate spectrophotometer, and urinary ZEN concentrations were calculated using RIDA ^®^SOFT Win (R-Biopharm, Art No. Z9999). Urinary Cre concentrations were determined using a commercial kit (Sikarikit-S CRE, Kanto Chemical, Tokyo, Japan) and a Hitachi 7700 clinical autoanalyzer (Hitachi High-Tech, Tokyo, Japan). ZEN concentrations were normalized to urinary Cre (pg/mg Cre) as previously described [3].

Serum AMH concentrations were measured using an ELISA kit (Ansh Labs, Webster, TX, USA) with 50 µL of undiluted serum. The assay’s limit of detection was 11 pg/mL, and the coefficient of variation was 2.9%. SAA concentrations were measured using an automated biochemical analyzer (Penta C200, HORIBA ABX SAS, Montpellier, France) with a reagent specialized for animal serum or plasma (VET-SAA ‘Eiken’ reagent; Eiken Chemical Co., Ltd., Tokyo, Japan). Final concentrations were calculated using a standard curve generated with the VET-SAA calibrator set (Eiken Chemical Co., Ltd., Tokyo, Japan).

### 5.7. Biochemical Analysis for Metabolic Profile Evaluation

The following concentrations were measured using the Labospect 7080 autoanalyzer (Hitachi, Tokyo, Japan) to evaluate metabolic profiles and conduct biochemical analyses: GOT, GGT, FFA, T-Cho, BUN, Glu, Ca, IP, Mg, TG, Vit. A, Vit. E, 3HB, TP, and Alb, following previously established protocols [49]. Blood serum was collected within 40 days of DFA III supplementation to monitor hepatic (GOT and GGT) and renal function (BUN), nutritional status (T-Cho, Glu, TG, FFA, Vit. A, Vit. E, 3HB, TP, and Alb), and mineral intake (Ca, IP, and Mg).

### 5.8. Measurement of SCFA with GC/MS

Serum SCFA concentrations were analyzed as previously reported [27]. Briefly, 1 mL of plasma was added to a 2 mL polypropylene tube containing a zirconia ball and 1 mL of 10% isobutyl alcohol. The sample was homogenized for 6 min using a TissueLyser II (Qiagen, Tokyo, Japan). After 5 min of centrifuging (21,000× *g*) the sample, 675 µL of the supernatant was transferred to a clean polypropylene tube. An internal standard (3-methylpentanoate) was added, followed by 400 µL of chloroform and 125 µL of 20 mM NaOH. After vortex mixing and centrifugation, the upper layer was collected and transferred to a glass vial. Derivatization was performed by adding 80 mL of isobutyl alcohol, 100 μL of pyridine, 70 μL of milli-Q water, and a boiling chip to the vial. Additionally, 50 μL of isobutyl chloroformate was added, and the mixture was vortexed. After derivatization, 150 μL of hexane was added, and the mixture was again vortexed and centrifuged at 21,000× *g*. The upper organic layer was collected for analysis, and 1 μL injected into GC/MS.

The derivatized SCFAs were measured using a GC-2030 gas chromatograph equipped with a QP2020 NX mass spectrometer and AOC-20i Plus autosampler (Shimadzu Corp., Kyoto, Japan). Separation was performed on a DB-5MS column (30 m length, 0.25 mm id, 0.25 µm film thickness; Agilent Technologies, Tokyo, Japan). The injector and detector temperatures were set at 260 °C and 280 °C, respectively. The column oven was programmed to heat from 40 °C (5 min) to 180 °C at a rate of 10 °C/min, then to 310 °C at a rate of 30 °C/min, followed by a final hold. Then, SCFA concentrations were analyzed by both selective ion monitor (SIM) and scan modes. Extracted ion chromatograms for individual SCFAs in SIM of GC/MS measurements are shown in Appendix A.

### 5.9. Statistical Analysis

A time-series analysis (days 0–40) was conducted for biochemical and SCFA data. As the results did not follow a log-normal distribution, natural log-transformation (ln) values were used. A linear mixed-effects model was employed, with each parameter set as a dependent variable. Measurement time points (0, 20, and 40 days) were treated as fixed effects, and individual cattle as random effects. Log-transformed estimates (mean and 95% CI) were back-transformed to obtain values on the original scale, representing the rate of change. Statistical comparisons were conducted using the mixed-effects model to determine whether significant changes occurred on days 20 and 40 relative to day 0. The Bonferroni correction was used to adjust for multiple comparisons. The two-sided *p* < 0.05 was considered statistically significant. All statistical analyses were performed using SPSS version 29.0 (IBM, Japan).

## Figures and Tables

**Figure 1 toxins-17-00409-f001:**
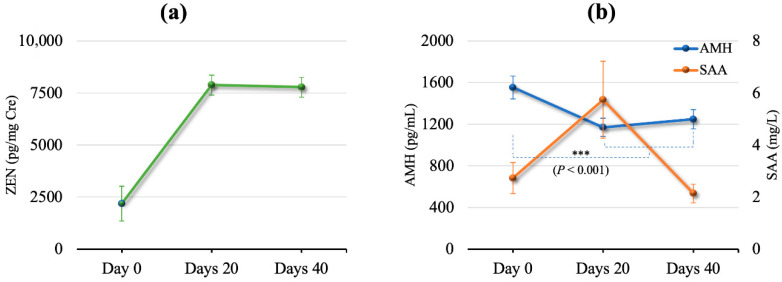
(**a**) ZEN/Cre concentration (mean ± SEM) in urine sample. (**b**) AMH and SAA concentrations in blood samples obtained on days 0, 20, and 40 after DFA III supplementation. *** indicates a significant difference (*p* < 0.001).

**Figure 2 toxins-17-00409-f002:**
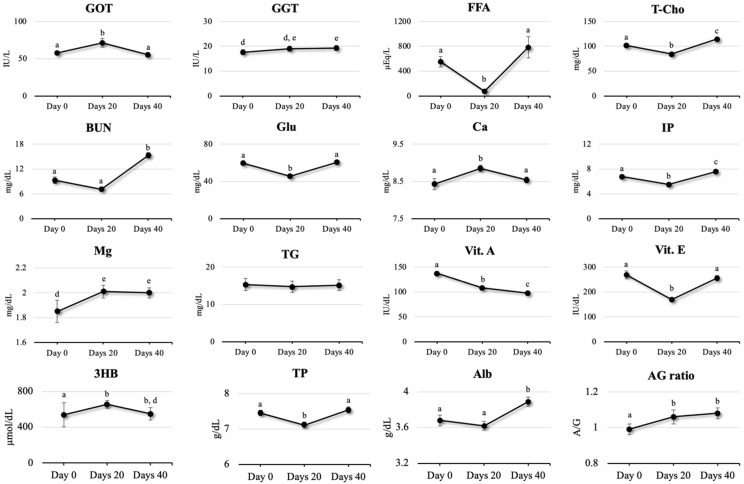
Nutritional condition (mean ± SEM) on days 0, 20, and 40 after difructose anhydride III supplementation. Hepatic function was assessed using glutamic oxaloacetic transaminase (GOT) and γ-glutamyl transferase (GGT). The nutritional condition was evaluated by determining the concentrations of free fatty acid (FFA), total cholesterol (T-Cho), blood urea nitrogen (BUN), glucose (Glu), triglyceride (TG), albumin (Alb), 3-hydroxybutyrate (3HB), and total protein (TP). Vitamin intake was determined using vitamin A (Vit. A) and vitamin E (Vit. E) concentrations. The mineral status was evaluated by determining the concentrations of calcium (Ca), inorganic phosphate (IP), and magnesium (Mg). The albumin-to-globulin (Alb: Glo) ratio was also determined. a–b; a–c, b–c indicates *p* < 0.01 and a–d; d–e indicates *p* < 0.05.

**Figure 3 toxins-17-00409-f003:**
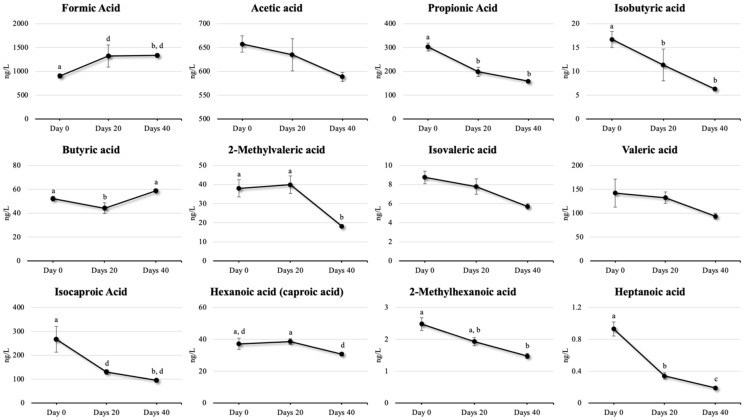
Concentrations of short-chain fatty acids (SCFAs) (mean ± SEM) on days 0, 20, and 40 after difructose anhydride III supplementation. a–b; a–c, b–c indicates *p* < 0.01 and a–d indicates *p* < 0.05.

**Figure 4 toxins-17-00409-f004:**
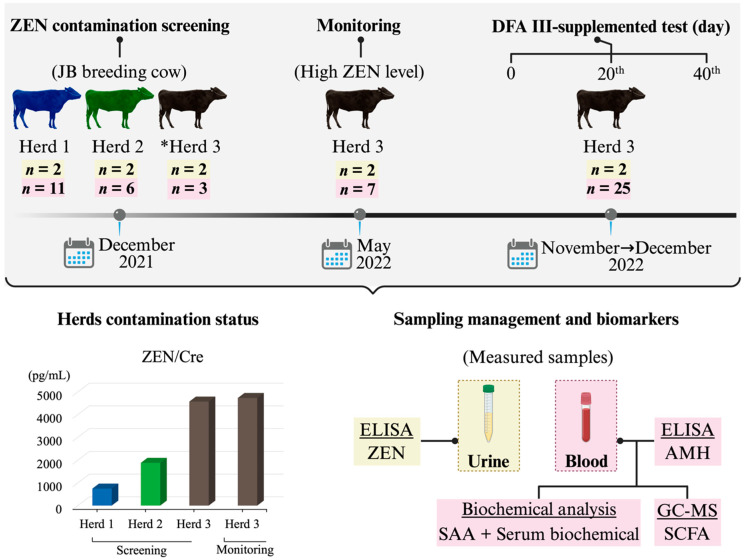
Schematic representation of the experimental design, sampling history, biomarkers, and ZEN contamination status in this study. *: High levels of urinary ZEN contamination were detected; →: Indicates the period between months; ZEN: zearalenone; Cre: Creatinine; JB: Japanese Black; DFA III: difructose anhydride III; AMH: anti-Müllerian Hormone; SAA: serum amyloid A; GC-MS: gas chromatography–mass spectrometry; SCFA: short-chain fatty acid.

## Data Availability

The original contribution presented in this study is included in the article/Appendix A. Further inquiries can be directed to the corresponding author.

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
