# Peer review of "Effect of Dietary Difructose Anhydride III Supplementation on the Metabolic Profile of Japanese Black Breeding Herds with Low-Level Chronic Exposure to Zearalenone in the Dietary Feed"

_toxins, 2025, doi:10.3390/toxins17080409_

Round 1
Reviewer 1 Report
Comments and Suggestions for Authors
Journal: Toxins
Manuscript ID: toxins-3745173
Title: Effect of dietary difructose anhydride III supplementation on the metabolic profile of Japanese Black breeding herds with low-level chronic exposure to zearalenone in the dietary feed
Comments to the Author
This manuscript addresses an important and timely issue concerning the mitigation of zearalenone contamination in cattle through dietary supplementation with difructose anhydride III (DFA III). The study is well-organized, presents a comprehensive dataset, and offers insights into the effects of DFA III on metabolic profiles, short-chain fatty acids (SCFAs), and biomarkers such as AMH and SAA in Japanese Black (JB) cattle. The experimental approach is generally sound, the experimental designs are no major flaws. While the study provides useful data and a compelling rationale for DFA III supplementation in the context of ZEN exposure and the results from this research would be informative for animal health and food safety. However, some minor points appear in the manuscript to amend before it can be acceptable for publication.
Discussion part
- The significant decrease in AMH is concerning, especially since ZEN is a known endocrine disruptor. It is unclear whether the DFA III supplementation helped protect ovarian function. Therefore, more information and a deeper discussion is needed to reconcile this apparent contradiction, particularly in relation to the proposed mechanism (intestinal barrier protection).
- The SCFA profile changes (e.g., transient decrease in butyrate) are not fully explained. Since SCFAs are critical indicators of gut health, the rationale behind their fluctuations post-DFA III supplementation needs better interpretation. The author should provide more information in the discussion to include possible shifts in microbial populations that might account for this, or propose future metagenomics studies.
Author Response
Reviewer 1
Comments and Suggestions for Authors
- The significant decrease in AMH is concerning, especially since ZEN is a known endocrine disruptor. It is unclear whether the DFA III supplementation helped protect ovarian function. Therefore, more information and a deeper discussion is needed to reconcile this apparent contradiction, particularly in relation to the proposed mechanism (intestinal barrier protection)
Thank you for this insightful comment. We agree with your remarks that it is unclear based on the results obtained from present study alone, whether the effects of intestinal barrier protection of DFA III helped against the effects of ZEN exposure or not. On the other hand, it is speculated that the intestinal barrier function induced by DFA III supplementation is exerted after the inflammatory state of the intestinal epithelial cells is improved, so it is possible that the effect may be seen with a time lag after the start of DFA III supplementation, which may have led to the inconsistency you pointed out. In any case, we agree that further research is needed into the mechanism of the effect of DFA III supplementation. Accordingly, we had added sentences regarding the points discussed above as follows: “On the other hand, it was unclear based on the results obtained from present study alone, whether the effects of intestinal barrier protection of DFA III helped against the effects of ZEN exposure or not. It was speculated that the intestinal barrier function induced by DFA III supplementation was exerted after the inflammatory state of the intestinal epithelial cells was improved, so it is possible that the effect may be seen with a time lag after the start of DFA III supplementation. Obviously, further research is needed to understand the mechanism of the effect of DFA III supplementation.” Lines 292-299.
- The SCFA profile changes (e.g., transient decrease in butyrate) are not fully explained. Since SCFAs are critical indicators of gut health, the rationale behind their fluctuations post-DFA III supplementation needs better interpretation. The author should provide more information in the discussion to include possible shifts in microbial populations that might account for this, or propose future metagenomics studies.
Thank you for your valuable opinions and suggestions. We have determined that the discussion of intestinal bacteria is inappropriate based on only the results of this study, and have therefore added that comprehensive further metagenomics studies are necessary in the future to clarify whether changes in the types and numbers of intestinal bacteria caused by the addition of DFA III result in changes in SCFA concentrations, as we have mentioned in the discussion part “Future research should investigate shifts in gut microbiota composition and population to better understand the fermentation behavior and metagenomic of DFA III in the intestinal tract.” Thank you very much for your kind understanding. Lines 356-358.

Reviewer 2 Report
Comments and Suggestions for Authors
The paper of “Effect of Dietary Difructose Anhydride III Supplementation on the Metabolic Profile of Japanese Black Breeding Herds with Low-Level Chronic Exposure to Zearalenone in the Dietary Feed” reported the evaluation of difructose anhydride III supplementation on the ZEN metabolic in the herd body. The study is valuable. However, there are several major problems in this manuscript, they need to be revised before publication.
(1) Original data of GC-MS spectra should be provided.
(2) Conclusion section is absent. And the collusion is not obvious, please clearly
(3) References should be cut half amount at least.
Author Response
Reviewer 2
- Original data of GC-MS spectra should be provided.
Thank you for this remark. According to the remark, we have added the extracted ion chromatogram and chromatograms of individual SCFAs in standard solution as Supplementary Table 5 and Supplementary Figure 1, respectively, and have added accompanying explanatory sentences in the main text. Lines 184-186, 507-510.
- Conclusion section is absent. And the collusion is not obvious, please clearly
Thank you for the suggestion. We have revised a conclusion for sentence as follows: “In conclusion, 40-day DFA III supplementation in cattle herds chronically exposed to low-level ZEN contamination may cover liver function, increased of nutritional status and immune response, and affect the blood SCFA concentration trends. DFA III as prebiotics, have beneficial effects on the integrity of the intestinal barrier and intestinal health. These findings suggested that DFA III supplementation could help mitigate the adverse effect of dietary ZEN contamination. Further accumulation and integration of field data are warranted to establish this strategy as a viable mycotoxin control approach in livestock farming”. Lines 371-379.
- References should be cut half amount at least.
Thank you for this remark. According to the remark, to reduce the number of references (especially self-citations) to 16%, we had removed 31 references from the original total citations. Thank you very much for your kind understanding.

Reviewer 3 Report
Comments and Suggestions for Authors
1) No where can I find how much DFM was fed.
2) The design is not clear why only 2 cows for urine?
3) all rpm needs to be converted to x g
4) Many statements throughout need references. If it is not common knowledge - it needs a reference
5) Please work with an English translation service.
Comments on the Quality of English Language
Please work with a translation service.
Author Response
Reviewer 3
- No where can I find how much DFA III was fed.
Thank you for this remark, and we are sorry for this careless mistake. Accordingly, we have added the information about amount of DFA III in Materials and Methods. Lines 417-419.
- The design is not clear why only 2 cows for urine?
Thank you for this comment. In our previous studies (Widodo et al., 2023; Widodo et al., 2024), although the number of cattle examined per herd was insufficient for statistical evaluation, we confirmed the validity of the feed analysis, and reported that it was possible to estimate the ZEN level in feed in a cattle herd by collecting at least two urine samples from cattle herds raised under the same feeding conditions. We are confident that our established urinary ZEN monitoring system in cattle herds will provide more important information on the ZEN contamination status in clinical settings of cattle. Thank you for your understanding.
- all rpm needs to be converted to x g
Thank you for this remark. Accordingly, we have changed the units used for centrifugation. Lines 449, 451, 462.
- Many statements throughout need references. If it is not common knowledge - it needs a reference
Thank you for this important comment regarding paper citations. On the other hand, other referees have asked to reduce the number of reference papers, including our own citations, by about half. Therefore, during our revision process of this manuscript, we have made revisions aimed at appropriate reference citations. Thank you very much for your understanding.
- Please work with an English translation service.
Thank you for bringing this to our attention. We apologize for the inconvenience. The revised version of the manuscript has been professionally edited by Editage and has been revised to improve the quality of the manuscript. We appreciate to your kind understanding.

Reviewer 4 Report
Comments and Suggestions for Authors
C1 - The abstract fails to mention the term "low level chronic" which is included in the title.
C2 - References in the text shall be marked with numbers placed in square brackets before the punctuation mark.
C3 - It should use the specific format of journal
The reference list should conform to the journal's requirements.
Author Response
Reviewer 4
- The abstract fails to mention the term "low level chronic" which is included in the title.
Thank you for this insightful comment. We have added the sentence “low-level chronic” in the abstract section accordingly. Line 11.
- References in the text shall be marked with numbers placed in square brackets before the punctuation mark.
Thank you for this remark. We have revised all references with numbers in square brackets to follow the format of the journal.
- It should use the specific format of journal
Thank you for this comment. Accordingly, the revised manuscript was edited according to the format specified by the journal.
- The reference list should conform to the journal's requirements.
According to the suggestion, we revised the reference list following the format of a journal.

Round 2
Reviewer 3 Report
Comments and Suggestions for Authors
Much improved manuscript.
One comment
line 44 what do you mean "and reticulum compartment unchanged" ?
Author Response
Toxins
Manuscript ID: toxins-3745173
Title: Effect of Dietary Difructose Anhydride III Supplementation on the Metabolic Profile of Japanese Black Breeding Herds with Low-Level Chronic Exposure to Zearalenone in the Dietary Feed
We thank Editor and all the Reviewers for their constructive comments. We have revised the manuscript in accordance with Reviewers’ suggestions. The revised sections of the manuscript and responses to the Reviewer 3 is marked in red for the revisions made in response to the comments of Reviewer 3. The revised manuscript has also undergone professional English editing (Editage) to improve its quality. Additionally, all elements of the statistical analysis described in the manuscript have undergone professional statistical analysis evaluation/review (Satista, Kyoto, Japan).
Reviewer 3
line 44 what do you mean "and reticulum compartment unchanged" ?
Thank you for pointing this out. We apologize for any confusion caused by this careless mistake. After reconfirming the context and deleting the section you pointed out, we have confirmed that the correct meaning is conveyed and have revised the relevant section. Lines 43-45.
